# Variability of Alternaria Leaf Spot Resistance in Jerusalem Artichoke (*Helianthus tuberosus* L.) Accessions Grown in a Humid Tropical Region

**Wanalai Viriyasuthee** [1], **Weerasak Saksirirat** [2], **Suwita Saepaisan** [2], **Mark L. Gleason** [3] **and Sanun Jogloy** [1,4,*]

[1] Department of Agronomy, Faculty of Agriculture, Khon Kaen University, Khon Kaen 40002, Thailand; wanalai_v@hotmail.com
[2] Department of Entomology and Plant Pathology, Faculty of Agriculture, Khon Kaen University, Khon Kaen 40002, Thailand; weerasak@kku.ac.th (W.S.); suwitsa@kku.ac.th (S.S.)
[3] Department of Plant Pathology and Microbiology, Iowa State University, Ames, IA 50011, USA; mgleason@iastate.edu
[4] Peanut and Jerusalem Artichoke Improvement for Functional Food Research Group, Department of Agronomy, Faculty of Agriculture, Khon Kaen University, Khon Kaen 40002, Thailand
[*] Correspondence: sanjogloy@gmail.com; Tel.: +66-43-202-209

**Abstract:** Alternaria leaf spot is an emerging disease of Jerusalem artichoke (*Helianthus tuberosus* L.) in tropical regions. The lack of known resistant germplasm sources is an important constraint to development of Jerusalem artichoke varieties with resistance to Alternaria leaf spot. The objectives of this study were to identify variability of Jerusalem artichoke genotypes for resistance to Alternaria leaf spot under field conditions and to investigate the relationships among resistance characters, yield, and yield components for selection of resistant varieties. Ninety six accessions of Jerusalem artichoke were evaluated in replicated trials under field conditions in early rainy and late rainy seasons in Khon Kaen, Thailand during 2014. Parameters evaluated included disease incidence, disease score, disease severity index, area under disease progress curve of disease incidence, area under disease progress curve of disease severity index, number of tubers/plants, tuber size, and fresh tuber yield. The genotypes HEL 335, HEL 256, HEL 317, HEL 308, and JA 86 were identified as sources of leaf spot resistance in both seasons. These genotypes can be used as sources of leaf spot resistance for Jerusalem artichoke breeding programs. HEL 293 and HEL 246 showed susceptibility to leaf spot disease in both seasons and should be used as standard susceptible checks.

**Keywords:** *alternaria* sp.; diversity of sunchoke; disease resistance; germplasm

## 1. Introduction

Jerusalem artichoke (*Helianthus tuberosus* L.) was initially domesticated in the temperate region of North America [1]. It was important as a food crop like potato for native Americans and European settlers. The carbohydrate in its tubers, in the form of inulin, can be used as a raw material for health food products, animal feed, and bioethanol [2,3]. Jerusalem artichoke is currently grown in most parts of the world and it is successfully established as a food crop in tropical regions [4]. However, production of Jerusalem artichoke in the tropics faces severe yield loss caused mainly by drought [5], stem rot [6], and leaf spot diseases. Stem rot caused by *Sclerotium rolfsii* is an important disease of Jerusalem artichoke in tropical regions and yield losses as high as 60% have been estimated [7]. Leaf spot is an emerging disease of Jerusalem artichoke in tropical regions. The disease causes severe leaf damage, lowers photosynthesis, and can reduce yield by up to 80% in *H. annuus* [8].

Jerusalem artichoke in temperate regions was shown to be moderately resistant to Alternaria leaf blight and stem spot caused by *Alternaria helianthi* [9], and it was used as a source of resistance to Alternaria leaf blight and stem spot in sunflower [10]. Alternaria leaf spot on Jerusalem artichoke in Thailand appears as small yellow spots on leaves; the spots eventually turn brown and are surrounded by yellow haloes. Thereafter, the spots expand and coalesce. The leaves show leaf blight symptoms, and defoliation begins on mature leaves and spreads upward to younger leaves.

Methods for control of leaf spot incited by *Alternaria* species have been investigated in sunflower and many of other crops. The disease can be controlled by several methods such as the use of resistant varieties, chemical control by fungicide applications [11], and biological control [12]. However, the lack of known resistant germplasm sources is an important constraint to the development of Jerusalem artichoke varieties with resistance to Alternaria leaf spot. The objectives of this study were to identify genotype variability of Jerusalem artichoke genotypes for resistance to Alternaria leaf spot under field conditions and to investigate the relationships among resistance characters, yield, and yield components for selection of resistant varieties.

## 2. Materials and Methods

### 2.1. Experimental Design and Treatments

Ninety six accessions of Jerusalem artichoke were received from the North Central Regional Plant Introduction Station (NCRPIS), Ames, IA, USA, the Leibniz Institute of Plant Genetics and Crop Plant Research (IPK), Stadt, Seeland, Germany, the Plant Gene Resources of Canada (PGRC) Agriculture and Agri-Food Canada, Saskatoon, Sasketchewan, Canada, and a commercial variety from Khon Kaen University, Khon Kaen, Thailand (Table 1). These accessions were evaluated in a randomized complete block design (RCBD) with three replications in the early rainy season from March to June 2014 and the late rainy season from September to December 2014 at the experimental farm of the Faculty of Agriculture, Khon Kaen University.

### 2.2. Preparation of Plant Materials and Field Management

Soil was ploughed three times and leveled using a tractor. Tubers of Jerusalem artichoke were cut into small pieces with two to three active buds and incubated at room temperature (28 ± 2 °C) and 80% relative humidity for one week to facilitate germination. Water was applied initially to moisten the medium (charred rice husks). Germinated tuber pieces were transferred to plug trays with mixed medium (soil:charred rice husk (1:1)) for one week until each seedling had two leaves. Healthy seedlings were transplanted to the field experiment in two-row sub-plots that were 4 m long and 1 m wide. Spacing was 0.5 m between rows and 0.5 m between plants within rows. The experimental plot was bordered by susceptible Jerusalem artichoke cultivar Kantawan 50-4 as spreader plants to provide a secondary source of inoculum. Manual weeding was performed twice at one and two months after transplanting. Chemical fertilizers (15-15-15 of N-$P_2O_5$-$K_2O$) at the rate of 156.3 kg/ha were applied to the crop one month after transplanting. Mini-sprinkler irrigation was available as necessary to avoid drought stress. Inoculation occurred through natural infection.

### 2.3. Data Collection

#### 2.3.1. Soil Data

Soil was sampled from 10 randomly selected positions on a zigzag transect in the experimental field. The soil samples were collected at 0–30 cm depth by auger [13]. The soil samples were mixed thoroughly to assure uniformity and air-dried. The dry soil was then analyzed for physical and chemical properties including texture, pH, organic matter [14], total N [15], available P [16], exchangeable K, exchangeable Ca, and cation exchange capacity.

**Table 1.** Jerusalem artichoke genotypes, sources of origin, and genetic resources [a].

| Entry No. | Varieties | Name of Varieties | Origin | Genetic Resources | Entry No. | Varieties | Name of Varieties | Origin | Genetic Resources |
|---|---|---|---|---|---|---|---|---|---|
| 1 | JA 1 | 7305 | Canada | PGRC | 49 | HEL 248 | Rote Zonenkugel | Germany | IPK |
| 2 | JA 2 | 7306 | Canada | PGRC | 50 | HEL 253 | – | Unknown | IPK |
| 3 | JA 6 | 7310 | Canada | PGRC | 51 | HEL 256 | – | Unknown | IPK |
| 4 | JA 7 | 7312 | Canada | PGRC | 52 | HEL 257 | – | Unknown | IPK |
| 5 | JA 8 | 7512 | Canada | PGRC | 53 | HEL 265 | BT4 | Hungry | IPK |
| 6 | JA 9 | 7513 | Canada | PGRC | 54 | HEL 272 | D19-63-340 | France | IPK |
| 7 | JA 10 | HM Hybrid A | Canada | PGRC | 55 | HEL 278 | Voelkenroder Spindel | Unknown | IPK |
| 8 | JA 12 | HM Hybrid C | Canada | PGRC | 56 | HEL 280 | BS-83-22 | Unknown | IPK |
| 9 | JA 14 | HM-3 | Canada | PGRC | 57 | HEL 288 | RA1 | Poland | IPK |
| 10 | JA 15 | HM-5 | Canada | PGRC | 58 | HEL 293 | RA9 | Poland | IPK |
| 11 | JA 16 | HM-7 | Canada | PGRC | 59 | HEL 308 | – | Unknown | IPK |
| 12 | JA 18 | HM-9 | Canada | PGRC | 60 | HEL 316 | – | Unknown | IPK |
| 13 | JA 20 | HM-11 | Canada | PGRC | 61 | HEL 317 | – | Unknown | IPK |
| 14 | JA 23 | DHM-3 | Canada | PGRC | 62 | [JA 102 × JA 89]-8 | Kantawan 50-4 | Thailand | KKU |
| 15 | JA 35 | W-97 | Canada | PGRC | 63 | JA 19 | HM-10 | Canada | PGRC |
| 16 | JA 36 | W-106 | Canada | PGRC | 64 | JA 22 | HM-13 | Canada | PGRC |
| 17 | JA 46 | DHM-14-3 | Canada | PGRC | 65 | JA 27 | DHM-7 | Canada | PGRC |
| 18 | JA 47 | DHM-14-6 | Canada | PGRC | 66 | JA 49 | 7513A | Canada | PGRC |
| 19 | JA 58 | Intress | USSR | PGRC | 67 | JA 95 | NACHODKA | USSR | PGRC |
| 20 | JA 59 | Volzskij-2 | USSR | PGRC | 68 | JA 98 | 242-62 | France | PGRC |
| 21 | JA 60 | Jamcovskij krashyj | USSR | PGRC | 69 | JA 99 | 29-65 | France | PGRC |
| 22 | JA 71 | TUB-675 USD-ARS-SR | USA | PGRC | 70 | JA 107 | 83-001-2 (37 × 6) | Canada | PGRC |
| 23 | JA 72 | TUB-676 USD-ARS-SR | USA | PGRC | 71 | JA 111 | 83-001-6 (37 × 6) | Canada | PGRC |
| 24 | JA 76 | #4 | Canada | PGRC | 72 | JA 113 | 83-001-8 (37 × 6) | Canada | PGRC |
| 25 | JA 77 | #5 | Canada | PGRC | 73 | JA 116 | 83-001-11 (37 × 6) | Canada | PGRC |
| 26 | JA 93 | Leningraskii (NC10-65) | USSR | PGRC | 74 | JA 119 | 83-002-1 (69 × 6) | Canada | PGRC |
| 27 | JA 108 | 83-001-3 (37 × 6) | Canada | PGRC | 75 | JA 125 | 83-005-1 (39 × 40) | Canada | PGRC |
| 28 | JA 109 | 83-001-4 (37 × 6) | Canada | PGRC | 76 | JA 127 | 83-006-1 (40 × 39) | Canada | PGRC |
| 29 | JA 114 | 83-001-9 (37 × 6) | Canada | PGRC | 77 | JA 129 | 83-006-4 (40 × 39) | Canada | PGRC |
| 30 | JA 122 | 83-004-2 (6 × 20) | Canada | PGRC | 78 | JA 130 | 83-006-5 (40 × 39) | Canada | PGRC |
| 31 | JA 132 | 83-007-2 (69 × 3) | Canada | PGRC | 79 | JA 133 | 83-007-4 (69 × 3) | Canada | PGRC |
| 32 | KKU Ac 001 | – | Unknown | – | 80 | JA 134 | 83-007-5 (69 × 3) | Canada | PGRC |
| 33 | CN 52867 | PGR-2367 | USSR | PGRC | 81 | JA 135 | 83-008-1 (69 × 39) | Canada | PGRC |
| 34 | JA 37 | Comber | Canada | PGRC | 82 | JA 21 | HM-12 | Canada | PGRC |
| 35 | JA 38 | B.C. #1 | Canada | PGRC | 83 | JA 3 | 7307 | Canada | PGRC |
| 36 | JA 67 | Oregon White | USA | PGRC | 84 | JA 123 | 83-004-4 (6 × 20) | Canada | PGRC |
| 37 | JA 89 | Waldspindel | France | PGRC | 85 | JA 86 | 79-62 | France | PGRC |
| 38 | JA 102 | 073-87 | Germany | PGRC | 86 | HEL 68 | – | Unknown | PGRC |
| 39 | HEL 53 | – | Germany | IPK | 87 | JA 55 | – | USA | PGRC |
| 40 | HEL 61 | Tambovskij Krasnyi | Russian Federation | IPK | 88 | JA 81 | Violet De Rennes | France | PGRC |
| 41 | HEL 62 | Sachalinskij Krasnyi | Russian Federation | IPK | 89 | JA 4 | 7308 | Canada | PGRC |
| 42 | HEL 65 | Sejanec 19 | Russian Federation | IPK | 90 | JA 5 | 7309 | Canada | PGRC |
| 43 | HEL 69 | – | Unknown | IPK | 91 | JA 117 | 83-001-12 (37 × 6) | Canada | PGRC |
| 44 | HEL 231 | – | Germany | IPK | 92 | JA 61 | VADIM | USSR | PGRC |
| 45 | HEL 335 | – | Unknown | IPK | 93 | JA 11 | HM Hybrid B | Canada | PGRC |
| 46 | Ames 2729 | TUB-49 | South Dakota | NCRPIS | 94 | JA 97 | D19-63340 | France | PGRC |
| 47 | HEL 243 | Bianka | Germany | IPK | 95 | HEL 66 | Kievskij Belyj | Ukraine | PGRC |
| 48 | HEL 246 | – | Unknown | IPK | 96 | JA 120 | 83-003-1 (6 × 20) | Canada | PGRC |

NCRPIS the North Central Regional Plant Introduction, IPK the Leibniz Institute of Plant Genetics and Crop Plant Research of Germany, PGRC the Plant Gene Resource of Canada. [a] Kays and Nottingham [4].

### 2.3.2. Meteorological Conditions

Weather data for the two seasons was recorded daily from transplanting until crop harvest at a weather station on the experimental farm of the Faculty of Agriculture, Khon Kaen University, Khon Kaen, Thailand. The weather station is located 0.5 and 1 km from experimental fields used during the early rainy season and late rainy season, respectively. Data included; maximum daily temperature, minimum daily temperature, mean daily relative humidity, and amount of rainfall.

### 2.3.3. Disease Data

Disease development was rated 18 times at 3-day intervals from 31 to 82 days after transplanting. Sixteen plants per plot were rated individually for number of infected plants and disease score. Sampled of symptomatic leaves were transported to a laboratory and checked for sporulation using a light microscope. Disease score was assessed using the method described by Mayee and Datar [17] for Alternaria leaf blight, where 0 = leaves free from infection, 1 = small irregular spots covering the leaves, 3 = small irregular brown spots with concentric rings covering 1%–10% leaf area, 5 = lesions enlarging, irregular brown with concentric rings covering 11%–25% leaf area, 7 = lesions coalesce to form typical blight symptoms covering 26%–50% leaf area, and 9 = lesions coalesce to typical blight symptoms covering >51% leaf area.

Disease incidence (DI) was calculated as follows [18]

$$DI~(\%) = (\text{number of infected plants} \times 100)/\text{total number of plants} \tag{1}$$

Disease severity index (DSI) was calculated as follows [18]

$$
\begin{aligned}
DSI~(\%) = \Sigma[(\text{rating score} \times \text{number of plants in rating}) \\
\times 100]/(\text{total number of sampled plants} \times \text{highest rating})
\end{aligned}
\tag{2}
$$

Area under the disease progress curve (AUDPC) was calculated for disease incidence (AUDPC-DI) and disease severity index (AUDPC-DSI) over time from 31 to 82 days after transplanting using the formulae as follows [19]

$$AUDPC = \Sigma[(X_i + X_{i+3})/2] \times (t_{i+3} - t_i) \tag{3}$$

where $x_i$ is disease incidence or disease severity on day $i$, $x_{i+3}$ is disease incidence or disease severity on day $i + 3$, $t_i$ is disease incidence or disease severity assessment on day $i$, and $t_{i+3}$ is disease incidence or disease severity assessment on day $i$ and $i + 3$.

### 2.3.4. Yield and Yield Components

The plants in each plot, without border row plants, were harvested at maturity. Three plants in each plot were sampled randomly from harvested plants and used for determination of yield components (number of tubers/plant and tuber size). Number of tubers from three plants were counted and averaged for number of tubers/plant. Total fresh tuber from three plants was weighted then divided by number of tubers to obtain average tuber size. For tuber yield were determined from nine plants, weighted and averaged to get fresh weight of tuber per plant.

### 2.4. Statistical Analysis

Data for each season were analyzed according to a randomized complete block design and error variances between the two seasons were tested for homogeneity. F-test was used to test the ratio of greater and lower error variance of two seasons. If the ratio is not larger than three folds then, the error variances could be considered homogeneity [20]. Data sets that complied with homogeneity of variance were subjected to combined analysis of variance for both seasons using the following model [21].

$$Y_{ijk} = \mu + S_i + \varepsilon_{ik} + V_j + SV_{ij} + \varepsilon_{ijk} \tag{4}$$

where $Y_{ijk}$ is the measured observation on the *ijkth* experimental unit (plot), μ is the overall mean, $S_i$ is the effect of the *ith* season, $\varepsilon_{ik}$ is the effect of the *ith* block within season, $V_j$ is the effect of the *jth* variety, $SV_{ij}$ is the interaction t of the *ith* level of *S* with the *jth* level of *V*, and $\varepsilon_{ijk}$ is pool error.

The disease incidence, disease severity index, and disease score at 76 days after transplanting were selected and presented for disease resistance because the data showed the highest F-test and the lowest CV value. Genotypes were categorized as susceptible, moderately resistant, or resistant based on mean separation of disease incidence. Genotypes were categorized as high and low yield and yield components based on mean separation of tuber yield. Means were compared by Duncan's multiple range test (DMRT). All calculations were done using the computer software MSTAT-C [22]. Pearson correlation was computed to determine the relationship between leaf spot disease resistance traits of the tested genotypes and association of resistant traits and agronomic traits. Correlation was calculated by using the STATISTIX8 software program [23].

## 3. Results

### 3.1. Weather and Soil Data

In the early rainy season, the minimum and maximum daily temperatures were 20.0 °C and 41.5 °C, respectively, the accumulated rain during crop season was 326.1 mm, and the relative humidity ranged from 72% to 97% (Figure 1a). In the late rainy season, the minimum and maximum daily temperatures were 21.0 °C and 35.5 °C, respectively, the accumulated rain during crop season was 126.9 mm and the relative humidity ranged from 67% to 97% (Figure 1b).

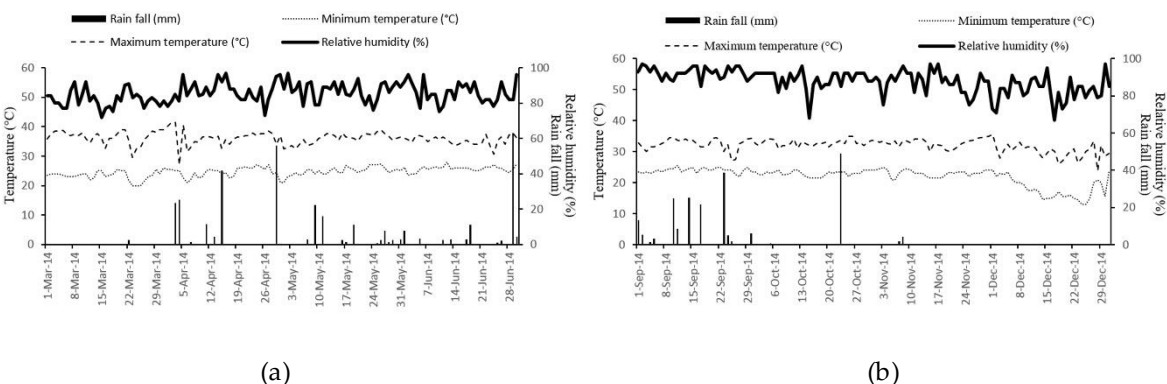

(a)                                      (b)

**Figure 1.** Rainfall (mm), relative humidity (%), maximum temperature (°C), and minimum temperature (°C) for the early rainy season (**a**) and the late rainy season (**b**).

The soil texture in the early rainy season was sand and in late rain season was loamy sand (data not shown). The soil chemical properties were as follows; pH 7.16 and 5.60, organic matter 0.432 and 0.467%, total nitrogen 0.021 and 0.023%, available phosphorus 28.07 and 33.48 mg/kg, available potassium 29.10 and 31.67 mg/kg, exchangeable calcium 455.81 and 360.00 mg/kg, electrical conductivity 0.054 and 0.022 dS/m, and cation exchange capacity (CEC) 5.245 and 1.433 c mol/kg for early rainy season and late rainy season, respectively.

### 3.2. Effect of Seasons and Varieties on Disease Parameters and Yield and Yield Components

Significant differences between the seasons, varieties and season by variety interaction were observed for disease incidence, disease severity index, disease score, AUDPC-DI, AUDPC-DSI, number of tubers per plant, tuber size, and tuber yield (Table 2).

**Table 2.** Mean squares for disease incidence (DI), disease severity index (DSI), disease score (Score), area under disease progress curve of disease incidence (AUDPC-DI), area under disease progress curve of disease severity index (AUDPC-DSI), number of tubers/plant, tuber size (TS), and tuber yield (TY) of 96 Jerusalem artichoke varieties in early rainy and late rainy seasons.

| SOV | Df | DI [a,b] | DSI [a,b] | Score [a,b] | AUDPC-DI [b] | AUDPC-DSI [b] | Tuber No./Plant | TS | TY |
|---|---|---|---|---|---|---|---|---|---|
| Season (S) | 1 | 113,906 * | 1429.41 ** | 91.76 ** | 16,170,000 * | 12,781.90 ** | 4230.42 * | 20,155.2 ** | 17,090,000 ** |
| Rep within season | 4 | 5556 | 19.97 | 1.13 | 1,233,817 | 121.90 | 128.13 | 2.6 | 31,960.1 |
| Varieties (V) | 95 | 3645 ** | 11.43 ** | 0.69 ** | 3,236,796 ** | 314.50 ** | 561.79 ** | 152.2 ** | 62,299.4 ** |
| S × V | 95 | 2503 ** | 6.3 ** | 0.38 ** | 964,196 ** | 79.30 ** | 446.31 ** | 123 ** | 34,019.8 ** |
| Pooled error | 380 | 1012 | 2.24 | 0.12 | 291,578 | 21.40 | 43.25 | 27.7 | 9203.66 |
| CV (%) | | 42.91 | 34.88 | 23.17 | 35.73 | 25.63 | 20.98 | 45.41 | 29.83 |

*, ** Significant at $p \leq 0.05$ and $p \leq 0.01$ respectively. [a] Disease incidence, disease severity index and score 76 days after transplanting in the early rainy and late rainy season. [b] Disease incidence, disease severity index, score, areas under disease progress curve of disease incidence and areas under disease progress curve of disease severity index transformed by square root.

Data of three disease resistance groups were selected for presentation (Tables 3 and 4). Disease incidence of testing entries in the early rainy season ranged from 0 to 100% with an average of 88.2% (Table 3) and ranged from 0 to 100% with the average of 59.7% in the late rainy season (Table 4). Disease severity index in the early rainy season ranged from 0 to 77.8% with the average of 38.7% ranged from 0 to 100% with the average of 11.1% in the late rainy season. Disease scores in the early rainy season ranged from 0 to 7 with an average of 3.5 and ranged from 0 to 9 with an average of 1.0 in the late rainy season. AUDPC-DI in the early rainy season ranged from 350 to 4571 with an average of 1679 and ranged from 0 to 3250 with an average of 1344 in the late rainy season. AUDPC-DSI in the early rainy season ranged from 150 to 2655 with an average of 580 and ranged from 0 to 3072 with an average of 274 in the late rainy season.

**Table 3.** Selected varieties of resistant, moderately resistant and susceptible Jerusalem artichoke evaluated in early rainy seasons.

| Groups | Entry | Varieties | DI (%) [a,b] | | DSI (%) [a,b] | | Score [a,b] | | AUDPC-DI [b] | | AUDPC-DSI [b] | |
|---|---|---|---|---|---|---|---|---|---|---|---|---|
| Resistant | 1 | HEL 335 | 0.0 | C | 0.0 | H | 0.0 | E | 600 | SU | 167 | Q-S |
| | 2 | JA 86 | 33.3 | BC | 18.5 | E-H | 1.7 | DE | 800 | Q-U | 322 | L-S |
| | 3 | HEL 256 | 33.3 | BC | 18.5 | E-H | 1.7 | DE | 650 | R-U | 194 | Q-S |
| | 4 | HEL 317 | 33.3 | BC | 3.7 | GH | 0.3 | E | 1183 | L-U | 331 | L-S |
| | 5 | JA 20 | 33.3 | BC | 11.1 | F-H | 1.0 | DE | 889 | O-U | 248 | P-S |
| | 6 | HEL 308 | 33.3 | BC | 18.5 | E-H | 1.7 | DE | 350 | U | 150 | S |
| Moderately resistant | 1 | JA 12 | 66.7 | AB | 37.0 | B-G | 3.3 | C-E | 1133 | L-U | 448 | G-S |
| | 2 | [JA 102 × JA 89] -8 | 66.7 | AB | 22.2 | C-H | 2.0 | C-E | 2528 | D-K | 599 | D-P |
| | 3 | HEL 280 | 66.7 | AB | 29.6 | B-G | 2.7 | C-E | 1081 | M-U | 346 | L-S |
| | 4 | JA 134 | 66.7 | AB | 29.6 | B-G | 2.7 | C-E | 1200 | K-U | 433 | H-J |
| | 5 | JA 98 | 66.7 | AB | 29.6 | B-G | 2.7 | C-E | 1617 | H-U | 613 | D-P |
| | 6 | JA 102 | 66.7 | AB | 37.0 | B-G | 3.3 | C-E | 567 | TU | 256 | N-S |
| | 7 | HEL 66 | 66.7 | AB | 37.0 | B-G | 3.3 | C-E | 567 | TU | 256 | N-S |
| Susceptible | 1 | JA 132 | 100.0 | A | 63.0 | A-C | 5.7 | A-C | 2433 | D-L | 949 | C-H |
| | 2 | JA 19 | 100.0 | A | 48.1 | A-E | 4.3 | B-D | 2271 | E-N | 968 | C-H |
| | 3 | HEL 288 | 100.0 | A | 40.7 | A-F | 3.7 | B-D | 3540 | A-D | 1119 | B-F |
| | 4 | JA 95 | 100.0 | A | 55.6 | A-D | 5.0 | B-D | 3286 | B-F | 1159 | B-E |
| | 5 | HEL 293 | 100.0 | A | 48.1 | A-E | 4.3 | B-D | 3467 | A-E | 1889 | B |
| | 6 | JA 2 | 100.0 | A | 55.6 | A-D | 5.0 | B-D | 3164 | B-G | 1196 | B-D |
| | 7 | HEL 246 | 100.0 | A | 77.8 | A | 7.0 | A | 4154 | AB | 2655 | A |
| | Min | | 0.0 | | 0.0 | | 0.0 | | 350 | | 150 | |
| | Max | | 100.0 | | 77.8 | | 7.0 | | 4571 | | 2655 | |
| | Mean | | 88.2 | | 38.7 | | 3.5 | | 1679 | | 580 | |
| | CV (%) | | 32.5 | | 30.8 | | 23.0 | | 38.2 | | 23.8 | |
| | F test | | 1.6 ** | | 2.3 ** | | 2.4 ** | | 5.1 ** | | 4.4 ** | |

** Significant at $p \leq 0.01$. Data are presented as minimum, maximum and mean values that were calculated from 96 varieties in early rainy season, values with different letters within the same column are significantly different at $p \leq 0.05$ by DMRT. [a] Disease incidence (DI), disease severity index (DSI) and disease scores at 76 days after transplanting in the early rainy and late rainy season. [b] Disease incidence, disease severity index, disease scores, areas under disease progress curve of disease incidence (AUDPC-DI) and areas under disease progress curve of disease severity index (AUDPC-DSI) were transformed by square root.

**Table 4.** Selected varieties of resistant, moderately resistant and susceptible Jerusalem artichoke evaluated in late rainy seasons.

| Groups | Entry | Varieties | DI (%) [a,b] | | DSI (%) [a,b] | | Score [a,b] | | AUDPC-DI [b] | | AUDPC-DSI [b] | |
|---|---|---|---|---|---|---|---|---|---|---|---|---|
| Resistant | 1 | JA 86 | 0.0 | B | 0.0 | G | 0.0 | G | 0 | Z | 0 | h |
| | 2 | HEL 256 | 0.0 | B | 0.0 | G | 0.0 | G | 100 | YZ | 11 | gh |
| | 3 | HEL 335 | 0.0 | B | 0.0 | G | 0.0 | G | 400 | U-Z | 44 | Z-h |
| | 4 | HEL 308 | 33.3 | AB | 3.7 | FG | 0.3 | FG | 450 | T-Z | 50 | V-h |
| | 5 | JA 15 | 33.3 | AB | 3.7 | FG | 0.3 | FG | 650 | P-Z | 74 | U-h |
| | 6 | HEL 317 | 33.3 | AB | 3.7 | FG | 0.3 | FG | 1050 | L-W | 117 | P-g |
| Moderately resistant | 1 | HEL 243 | 66.7 | AB | 7.4 | E-G | 0.7 | E-G | 600 | Q-Z | 67 | S-h |
| | 2 | HEL 316 | 66.7 | AB | 7.4 | E-F | 0.7 | E-G | 700 | O-Z | 78 | Q-g |
| | 3 | HEL 61 | 66.7 | AB | 7.4 | E-G | 0.7 | E-G | 1100 | L-W | 144 | O-f |
| | 4 | JA 20 | 66.7 | AB | 7.4 | E-G | 0.7 | E-G | 1500 | G-O | 167 | G-a |
| | 5 | JA 134 | 66.7 | AB | 7.4 | E-G | 0.7 | E-G | 1600 | F-N | 200 | G-W |
| | 6 | HEL 65 | 66.7 | AB | 7.4 | E-G | 0.7 | E-G | 1700 | F-N | 211 | F-V |
| | 7 | JA 113 | 66.7 | AB | 7.4 | E-G | 0.7 | E-G | 1700 | F-N | 278 | E-P |
| Susceptible | 1 | JA 5 | 100.0 | A | 33.3 | BC | 3.0 | BC | 2150 | C-J | 717 | CD |
| | 2 | JA 117 | 100.0 | A | 48.1 | B | 4.3 | B | 2550 | A-E | 983 | BC |
| | 3 | JA 95 | 100.0 | A | 33.3 | BC | 3.0 | BC | 2550 | A-E | 1094 | B |
| | 4 | JA 93 | 100.0 | A | 55.6 | B | 5.0 | B | 2450 | B-F | 1117 | B |
| | 5 | JA 109 | 100.0 | A | 55.6 | B | 5.0 | B | 3100 | AB | 1300 | B |
| | 6 | HEL 293 | 100.0 | A | 100.0 | A | 9.0 | A | 3100 | AB | 2889 | A |
| | 7 | HEL 246 | 100.0 | A | 100.0 | A | 9.0 | A | 3250 | A | 3072 | A |
| | | Min | 0.0 | | 0.0 | | 0.0 | | 0.0 | | 0 | |
| | | Max | 100.0 | | 100.0 | | 9.0 | | 3250 | | 3072 | |
| | | Mean | 59.7 | | 11.1 | | 1.0 | | 1344 | | 274 | |
| | | CV (%) | 57.3 | | 40.5 | | 20.5 | | 30.9 | | 27.7 | |
| | | F test | 4.2 ** | | 8.5 ** | | 11.2 ** | | 12.2 ** | | 19.4 ** | |

** Significant at $p \leq 0.01$. Data are presented as minimum, maximum and mean values that were calculated from 96 varieties in early rainy season, values with different letters within the same column are significantly different at $p \leq 0.05$ by DMRT. [a] Disease incidence (DI), disease severity index (DSI) and disease scores at 76 days after transplanting in the early rainy and late rainy season. [b] Disease incidence, disease severity index, disease scores, areas under disease progress curve of disease incidence (AUDPC-DI) and areas under disease progress curve of disease severity index (AUDPC-DSI) were transformed by square root.

In the early rainy season, Jerusalem artichoke accessions could be classified into distinct groups based on reaction to the disease. The selected resistant group included HEL 335, JA 86, HEL 256, HEL 317, JA 20, and HEL 308 and the susceptible group consisted of JA 132, JA 19, HEL 288, JA 95, HEL 293, JA2, and HEL 246 (Table 3). In the late rainy season, the selected resistant group comprised JA 86, HEL 256, HEL 335, HEL 308, JA 15, and HEL 317, and the susceptible group included JA 5, JA 117, JA 95, JA 93, JA 109, HEL 293, and HEL 246 (Table 4).

Five Jerusalem artichoke genotypes showed low disease parameters for both seasons HEL 335, HEL 256, HEL 317, HEL 308, and JA 86 (Tables 3 and 4).

For yield and yield components, the number of tubers/plants in the early rainy season ranged from 6 to 89 with an average of 29 (Table 5) and the number of tubers in the late rainy season ranged from 14 to 78 with an average of 34 (Table 6). Tuber size in the early rainy season ranged from 1.5 to 15.6 g/tuber with an average of 5.7 g/tuber and ranged from 3.6 to 44.1 g/tuber with an average of 17.5 g/tuber in the late rainy season. Tuber yield in the early rainy season ranged from 28.1 to 365.2 g/plant with an average of 149.3 g/plant, and tuber yields in the late rainy season ranged from 123.6 to 913.6 g/plant with an average of 493.9 g/plant.

In the early rainy season, JA 9, JA 8, JA 18, JA 116, JA 46, JA 27, JA 58, JA 49, JA 59, and JA 71 formed a group with low yield and yield components, whereas HEL 243, JA 134, JA 15, JA 6, HEL 280, HEL 257, JA 123, JA 122, HEL 278, and JA 95 formed a group with high yield and yield components (Table 5). In the late rainy season, JA 21, JA 76, JA 27, JA 35, JA 22, JA 6, JA 9, JA 49, JA 59, and JA 117 were classified as the group with low yield and yield components, whereas JA 129, JA 60, JA 111, JA 58, HEL 278, JA 102, JA 120, HEL 65, HEL 280, and JA 37 were classified as the group with high yield and yield components (Table 6).

**Table 5.** Selected varieties of yield and yield component Jerusalem artichoke evaluated in early rainy season.

| Groups | Entry | Varieties | Number of Tubers/Plant | | Tuber Size (g/Tuber) | | Tuber Yield (g/Plant) | |
|--------|-------|-----------|---------|------|--------|------|---------|------|
| Low | 1 | JA 9 | 14 | g-l | 2.0 | X-a | 28.1 | l |
| | 2 | JA 8 | 12 | i-l | 2.3 | W-a | 28.4 | l |
| | 3 | JA 18 | 14 | g-l | 2.5 | V-a | 33.7 | kl |
| | 4 | JA 116 | 21 | X-h | 1.7 | Za | 35.0 | j-l |
| | 5 | JA 46 | 13 | h-l | 2.9 | U-a | 35.8 | l-l |
| | 6 | JA 27 | 20 | X-h | 1.8 | Y-a | 37.5 | h-l |
| | 7 | JA 58 | 12 | j-l | 3.7 | Q-a | 38.1 | h-l |
| | 8 | JA 49 | 26 | P-b | 1.5 | a | 39.6 | h-l |
| | 9 | JA 59 | 32 | J-T | 2.2 | W-a | 48.2 | g-l |
| | 10 | JA 71 | 18 | a-j | 2.9 | U-a | 53.3 | f-l |
| High | 1 | HEL 243 | 25 | R-d | 10.3 | B-E | 252.7 | B-G |
| | 2 | JA 134 | 29 | M-Z | 9.0 | C-H | 257.1 | B-F |
| | 3 | JA 15 | 37 | G-N | 6.9 | E-S | 257.7 | B-F |
| | 4 | JA 6 | 44 | D-H | 5.9 | G-W | 257.9 | B-F |
| | 5 | HEL 280 | 51 | C-D | 5.2 | J-a | 263.0 | B-E |
| | 6 | HEL 257 | 39 | E-L | 7.2 | E-Q | 280.0 | B-D |
| | 7 | JA 123 | 36 | H-P | 7.8 | D-N | 282.4 | B-D |
| | 8 | JA 122 | 22 | V-h | 13.4 | AB | 289.7 | BC |
| | 9 | HEL 278 | 36 | H-Q | 8.5 | C-K | 300.3 | B |
| | 10 | JA 95 | 45 | D-G | 8.0 | D-L | 365.2 | A |
| | | Min | 6 | | 1.5 | | 28.1 | |
| | | Max | 89 | | 15.6 | | 365.2 | |
| | | Mean | 29 | | 5.7 | | 149.3 | |
| | | CV (%) | 17.1 | | 31.6 | | 23.4 | |
| | | F test | 22.8 ** | | 7.6 ** | | 13.7 ** | |

** Significant at $p \leq 0.01$. Data were presented minimum, maximum and mean values were calculated from 96 varieties in early rainy season, values with different letters within the same column are significantly different at $p \leq 0.05$ by DMRT.

**Table 6.** Selected varieties of yield and yield component Jerusalem artichoke evaluated in late rainy season.

| Groups | Entry | Varieties | Number of Tubers/Plant | | Tuber Size (g/Tuber) | | Tuber Yield (g/Plant) | |
|--------|-------|-----------|---------|------|--------|------|---------|------|
| Low | 1 | JA 21 | 18 | a-h | 10.0 | U-d | 123.6 | c |
| | 2 | JA 76 | 43 | E-R | 3.6 | d | 151.5 | bc |
| | 3 | JA 27 | 47 | C-L | 3.9 | d | 165.9 | a-c |
| | 4 | JA 35 | 16 | d-h | 15.5 | I-d | 248.1 | Z-c |
| | 5 | JA 22 | 15 | e-h | 17.5 | G-d | 256.0 | Y-c |
| | 6 | JA 6 | 60 | B-D | 4.7 | cd | 266.7 | X-c |
| | 7 | JA 9 | 22 | W-h | 12.2 | N-d | 268.7 | X-c |
| | 8 | JA 49 | 30 | M-h | 13.9 | K-d | 278.3 | W-c |
| | 9 | JA 59 | 28 | Q-h | 11.2 | Q-d | 292.1 | V-c |
| | 10 | JA 117 | 52 | C-H | 5.7 | b-d | 298.2 | V-c |
| High | 1 | JA 129 | 52 | C-G | 13.2 | L-d | 684.0 | A-I |
| | 2 | JA 60 | 34 | K-a | 27.2 | B-N | 701.9 | A-H |
| | 3 | JA 111 | 26 | S-h | 27.8 | B-L | 712.7 | A-G |
| | 4 | JA 58 | 29 | O-h | 25.4 | C-S | 733.0 | A-F |
| | 5 | HEL 278 | 23 | W-h | 32.5 | A-F | 745.8 | A-E |
| | 6 | JA 102 | 34 | J-a | 24.6 | D-V | 816.9 | A-D |
| | 7 | JA 120 | 22 | W-h | 40.3 | AB | 833.7 | A-C |
| | 8 | HEL 65 | 42 | E-T | 28.4 | B-K | 836.1 | A-C |
| | 9 | HEL 280 | 36 | H-Y | 24.7 | D-U | 861.0 | AB |
| | 10 | JA 37 | 34 | J-a | 26.4 | B-P | 913.6 | A |
| | | Min | 14 | | 3.6 | | 123.6 | |
| | | Max | 78 | | 44.1 | | 913.6 | |
| | | Mean | 34 | | 17.5 | | 493.9 | |
| | | CV (%) | 23.2 | | 41.3 | | 26.6 | |
| | | F test | 7.4 ** | | 4.8 ** | | 4.6 ** | |

** Significant at $p \leq 0.01$. Data were presented minimum, maximum and mean values were calculated from 96 varieties in early rainy season, values with different letters within the same column are significantly different at $p \leq 0.05$ by DMRT.

### 3.3. Correlation between Disease Parameters

The correlation coefficients among the parameters for leaf spot resistance were positive and highly significant in both seasons. In the early rainy season, high positive correlation coefficients were

found between AUDPC-DI and AUDPC-DSI (0.81**) whereas the rest of the correlation coefficients of disease parameters were moderately positive (Table 7). In the late rainy season, high correlation coefficients were found between disease incidence and AUDPC-DI (0.85**) and disease severity index and AUDPC-DSI (0.97**) (Table 8).

**Table 7.** Correlation coefficients of DI, DSI, AUDPC-DI and AUDPC-DSI, number of tuber/plant (No. of tuber), TS (g/tuber), and TY (g/plant) of 96 Jerusalem artichoke varieties in early rainy.

| Characters | DI | DSI | AUDPC-DI | AUDPC-DSI | No. of Tuber | TS |
|---|---|---|---|---|---|---|
| DSI | 0.66 ** | | | | | |
| AUDPC-DI | 0.41 ** | 0.37 ** | | | | |
| AUDPC-DSI | 0.38 ** | 0.63 ** | 0.81 ** | | | |
| No. of tuber | 0.10 | 0.22 | 0.14 | 0.26 * | | |
| TS | −0.17 | −0.16 | 0.16 | −0.01 | −0.28 ** | |
| TY | 0.06 | 0.15 | 0.25 * | 0.22 * | 0.53 ** | 0.56 ** |

** Significant at $p \leq 0.01$ probability level, * Significant at $p \leq 0.05$ probability level respectively.

**Table 8.** Correlation coefficients of DI, DSI, AUDPC-DI and AUDPC-DSI, number of tubers/plant (No. of tuber), TS (g/tuber), and TY (g/plant) of 96 Jerusalem artichoke varieties in late rainy season.

| Characters | DI | DSI | AUDPC-DI | AUDPC-DSI | No. of Tuber | TS |
|---|---|---|---|---|---|---|
| DSI | 0.52 ** | | | | | |
| AUDPC-DI | 0.85 ** | 0.67 ** | | | | |
| AUDPC-DSI | 0.42 ** | 0.97 ** | 0.65 ** | | | |
| No. of tuber | −0.14 | −0.04 | 0.02 | −0.02 | | |
| TS | 0.03 | −0.05 | −0.1 | −0.05 | −0.70 ** | |
| TY | −0.11 | −0.12 | −0.09 | −0.07 | −0.13 | 0.69 ** |

** Significant at $p \leq 0.01$ probability level, * Significant at $p \leq 0.05$ probability level respectively.

*3.4. Correlation between Disease Parameters and Yield and Yield Components*

In early rainy season, no correlation of disease resistance parameters with yield and yield components was found, except of AUDPC-DI with tuber yield (0.25*), AUDPC-DSI with number of tubers/plant (0.26*), and AUDPC-DSI with tuber yield (0.22*) (Table 7). In the late rainy season, no correlation of disease resistant parameters with yield and yield components was noted (Table 8).

*3.5. Correlation between Yield and Yield Components*

In the early rainy season, correlation coefficients among the yield and yield components were significantly positive. Tuber yield correlated with number of tubers/plant (0.53**) and tuber size (0.56**). Negative correlation has found between number of tubers/plant and tuber size (−0.28**) (Table 7). For late rainy season, we found a positive correlation between tuber yield and tuber size (0.69**) and a negative correlation between number of tubers per plant and tuber size (−0.70**) (Table 8).

**4. Discussion**

The studies of diversity in Jerusalem artichoke had been conducted for yield components [24], inulin content [25], morphological traits and agronomic traits [26,27], and stem rot resistance [28]. For leaf spot disease of *Helianthus* species, there was only one study in the temperate zone [9]. To our knowledge, genotypic resistance to Alternaria leaf spot in tropical area has not been reported previously in Jerusalem artichoke. Alternaria leaf spot can destroy Jerusalem artichoke leaves, which are the main source of photosynthesis, reducing photosynthetic area and yield. Leaf spot disease caused by *A. alternata* destroys the active leaf area and reduces yield of sunflower [29]. Sunflowers with disease severity higher than 10% yielded less than 500 kg/ha [30]. For mustard and rapeseed, leaf spot disease reduced yield up to 70% [31].

In the present study, genotypic diversity of Jerusalem artichoke for resistance to Alternaria leaf spot was highly significant among accessions and was arranged into three groups including resistant, moderately resistant, and susceptible accessions. The varieties were classified into different 3 groups in two seasons. HEL335, HEL256, HEL317, HEL308, and JA86 showed high level of resistance to leaf spot

disease in both seasons, whereas HEL 293 and HEL 246 showed susceptibility to leaf spot disease in both seasons. These groups of genotypes can be used as sources of resistance and standard susceptible checks, respectively, for leaf spot disease evaluations in breeding programs of Jerusalem artichoke. In sunflower, the genetic control of resistance to Alternaria leaf blight was polygenic and conferred by dominant genes [32].

In this study, all disease parameters in the early rainy season were higher than in the late rainy season (Tables 3 and 4). Season significantly affected disease incidence and disease severity index (Table 2). In the early rainy season, relative humidity was consistently higher than in the late rainy season. The range of relative humidity was 72%–97% in the early rainy season and 67%–97% in the late rainy season. In the late rainy season, during the critical time for disease development at 60 days after transplanting, relative humidity was lower than during the early rainy season. In the early rainy season, the AUDPC-DI and DSI was higher than in the late rainy season. In the early rainy season relative humidity was consistently higher throughout the testing season but in late rainy season, the relative humidity was lower after 60 days after transplanting. The relative humidity may be the main factor for conidia germination, leaf penetration, and development of the disease. Green and Bailey [33] found that *A. cirsinoxia* conidia germinated well under relative humidity higher than 90%. The temperature also may have affected disease progress. The temperature in the early rainy season (20–41.5 °C) was higher than in late rainy season (21–35.5 °C). The optimum temperature for germination of *Alternaria* is 24 °C in laboratory conditions. The influence of temperature on Alternaria blight development of sunflower also varied between crop and season [34]. Rainfall did not affect disease incidence and disease severity, possibly because the experiment was conducted under irrigation with a mini-sprinkler in both seasons. In sunflowers, development of Alternaria blight under field conditions was related to minimum and maximum temperature and relative humidity [32]. Not only weather parameters, but also plant physiological growth stage affected Alternaria blight development in mustard [35].

The results of combined analysis of variance shown highly significant of variety by season inter action for all disease parameters (Table 2) indicated that the performance of the tested genotypes for disease resistance was inconsistent across seasons. A similar report of screening of a potato for resistance to early blight showed low correlation between the seasons [36]. Therefore, screening of leaf spot disease resistance in Jerusalem artichoke for disease incidence and disease severity index should be conducted in at least two seasons.

Positive correlation was found between disease parameters in this experiment (Tables 7 and 8). In early rainy season, AUDPC-DI and AUDPC-DSI was very strongly correlated (Table 7) and in late rainy season, high correlations between AUDPC-DI with disease incidence and AUDPC-DSI with disease severity index were observed (Table 8). In Alternaria blight of sunflower [32], mustard [31], and *Brassica* [37], disease incidence and disease severity index were used as resistant indexes. Correlation between disease parameters could help breeders use alternative traits as indirect selection indexes for improving resistant genotypes of Jerusalem artichoke.

In this study, Jerusalem artichoke grown in the early rainy season had lower yield and yield components than did the crop grown in the late rainy season. Season was the main source of variation in number of tubers per plant, tuber size, and tuber yield. The variations in these traits as affected by seasonal variations would be due to the fact that quantitative traits are controlled by multiple genes with combined effect, and expression of these traits can vary greatly depending on environment [38]. Several quantitative traits such as tuber yield, tuber size, inulin content, and maturity are economically important [25]. HEL278 and HEL280 had the highest yield and yield components in both seasons. HEL 278 showed susceptibility to leaf spot disease whereas HEL 280 showed moderate resistance to disease. These genotypes can be used as sources for breeding programs to improve yield and yield components in Jerusalem artichoke. In general, no significant correlation was found between disease parameters and yield and yield components in both seasons. The results indicated that selection for high yield and desirable yield components with Alternaria resistance is possible with the tested

materials. It is possible that severity of Alternaria leaf spot would need to be higher than in our study in order to increase yield loss.

## 5. Conclusions

In conclusion, variation of Jerusalem artichoke genotypes for Alternaria leaf spot was grouped into three groups including resistant, moderately resistant, and susceptible. HEL335, HEL256, HEL317, HEL308, and JA86 were resistant genotypes and HEL 293 and HEL 246 were classified to susceptible genotypes. These groups can be used as sources of resistance and susceptible check, respectively, for breeding of leaf spot disease resistance. HEL278 and HEL280 had the highest yield and yield components in both seasons. These genotypes can be used as sources for breeding programs to improve yield and yield components in Jerusalem artichoke. Selection of Jerusalem artichoke for high yield and desirable yield components with Alternaria resistance is possible because of no correlation between agronomic traits with leaf spot disease resistance.

**Author Contributions:** Conceptualization, W.S., S.S. and S.J.; Data curation, W.V.; Formal analysis, W.V.; Investigation, W.V.; Methodology, W.S., S.S. and S.J.; Resources, M.G.; Supervision, S.J.; Writing—original draft, W.V.; Writing—review and editing, W.V., M.G. and S.J.

**Funding:** This research was funded by the Royal Golden Jubilee Ph.D. Program (grant no. PHD/0066/2556), the Thailand Research Fund through the Senior Scholar Project of Sanun Jogloy (RTA6180002) and Peanut and Jerusalem Artichoke improvement for Functional Food Project, Khon Kaen University.

**Acknowledgments:** The Thailand Research Fund through the Senior Scholar Project of Sanun Jogloy (RTA6180002) and Peanut and Jerusalem Artichoke improvement for Functional Food Project, Khon Kaen University. The North Central Regional Plant Introduction Station (NCRPIS), Ames, IA, USA, the Leibniz Institute of Plant Genetics and Crop Plant Research (IPK), Stadt, Seeland, Germany, and the Plant Gene Resources of Canada (PGRC) Agriculture and Agri-Food Canada, Saskatoon, Sasketchewan, Canada are acknowledged for their donation of Jerusalem artichoke germplasm.

**Conflicts of Interest:** The authors declare no conflict of interest.

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
