# Peer review of "Variability of Alternaria Leaf Spot Resistance in Jerusalem Artichoke (Helianthus tuberosus L.) Accessions Grown in a Humid Tropical Region"

_agronomy, doi:10.3390/agronomy9060268_

Round 1
Reviewer 1 Report
Viriyasuthee et al. present a manuscript on resistance to Alternaria in Helianthus tuberosus with the aim to identified resistant accessions for breeding of successfully this tuber crop in tropical environment. They found that the severity of Alternaria depend on the season and the genotypes, therefore GxE. They also show that the 96 accessions they studied contain genetic diversity for resistance and yield.
This article is novel the study of traits in H. tuberosus in tropical environment. The genetic basis of tuber size and number has been published recently and strongly depend on heterozygosity in this polyploid species.
As general information to the author, some of the H. tuberosus accessions between and within collections (PGRC, IPK) are clones. See Bock et al 2018 in Nature Ecol&Evol, Fig S7. The genotype identified as resistant and susceptible may be single clones. You can also potentially make use of the genomic data from that study, freely available on genbank/Sequence read archive for most of your accessions.
Overall, the author could make better use of their data and need to complete the description of the statistical method used.
Main comments:
- Statistical analyses are not well described. I assumed that a linear model and anova was performed for table 2. The details of the model should be specified. For example, was the replicates nested within season for the analysis? Which correlation measurement (Spearman, Pearson?) was performed for table 7 and 8?
- I had difficulties understanding what the AUDPC values are. Maybe describe it as disease severity over time? Also, when where the disease incidence and disease severity measured? It’s a single data point, right? Is it at day 30, day 83 or sometime in between?
- Fig 2A: Doing a regression on categorical data is quite weak in power. A rank based correlation analysis would be better for here. Also why only 12 points when you have 96 accessions?
-Fig. 2 in general: Maybe a multi-phenotype (including yield and resistance) PCA would help you visualize your accession in the phenotypic space and identify potential clones in addition to visualize your data across season.
Fig. 2: Does the correlation between early and late season aim to show the effect of the genotypes on your traits across season? You already showed that in the table 2. Plotting the phenotypes in different colors for the season and showing the effect of the season on the regressions would be more interesting here.
Line 322: “Positive correlation was found for all traits in this experiment”. This claim is totally wrong. Tuber number and tuber size are negatively correlated for example.
Line 323: “Disease severity and disease scores were very strongly correlated”. This is an empty claim. I would worry if they were not, knowing that they are calculated based on the same phenotype, e.g. disease score.
Author Response
Dear Editor/Reviewers
We are pleased to resubmit the manuscript no Agronomy-489691 entitled "Variability of Alternaria leaf spot resistance in Jerusalem artichoke (Helianthus tuberosus L.) accessions grown in a humid tropical region" after revision for re-evaluation and possible publication in the Agronomy.
The authors would like to thank the editor and reviewers for all valuable comments and suggestion on the manuscript in order to improve the manuscript.
We did revise the manuscript base on editor and reviewers' comments and suggestions all of the points and responses to the editor and reviewers of each point had been done and attached with this file.
Best Regards,
Sanun Jogloy

Reviewer 2 Report
The manuscript is generally well written, although the topic is not particularly novel. However, the results add up to the literature on screening crop genotypes for resistance to Alternaria leaf spot. The following items need to be addressed by the authors in order to improve the manuscript and make it suitable for publication.
1) Disease assessment data. The disease severity (DS) as used by the authors is actually Disease Severity Index (DSI), which incorporates disease rating score and number of samples in each class of the scoring scale. The scoring scale here is from 0 to 9, and is based on disease severity (which is the extent of symptom expression on each leaf). The authors need to clarify this difference.
2) Statistical analysis (homogeneity tests). The authors stated that “data for each season were analyzed according to a randomized complete block design and error variances between the two seasons were tested for homogeneity.” The authors need to provide details on how homogeneity tests were specifically conducted.
3) Statistical analysis (correlation analysis). The authors need to specify which type of correlation analysis was used. Typically, if they are using Pearson correlation, the variables need to be continuous. However, Score data are ordinal, and not continuous data type. In this case, Spearman correlation analysis may be more appropriate. So, the authors need to reexamine data in Tables 7 and 8, particularly the correlation between Scores and other variables.
4). The range of the data presented in Figure 2e does not match the data type, which is the Score data. Indeed, the Score data spans from 0 to 9. So, all data should fall within this range and not between 0 and 100 or 0 and 140. The authors need to check on this figure for accuracy of the data presented.
Other suggestions/comments
Line 51. Please clarify the use of “Alternaria sp.” Are the authors referring to an unknown species of Alternaria (in which case, “sp.” can be used after Alternaria) or several special species of Alternaria (in which case “spp.” should be used after Alternaria)
Lines 102-103. Change “Disease score was assessed according to described by Mayee and Datar” TO “Disease score was assessed using the method described by Mayee and Datar”
Lines 122-125. The authors stated that: “Fresh tuber weight from three plants was determined then divided by number of tubers to obtain average tuber size. For tuber fresh weight were determined from nine plants, weighed and averaged to get fresh weight of tuber per plant.” There is inconsistency here. In the first sentence, the authors indicate that tuber weight was assessed from “three plants” and in the second sentence, they mentioned “nine plants.” Which is the correct number of plants?
Author Response
April 27, 2019
Dear Editor/Reviewers
We are pleased to resubmit the manuscript no Agronomy-489691 entitled "Variability of Alternaria leaf spot resistance in Jerusalem artichoke (Helianthus tuberosus L.) accessions grown in a humid tropical region" after revision for re-evaluation and possible publication in the Agronomy.
The authors would like to thank the editor and reviewers for all valuable comments and suggestion on the manuscript in order to improve the manuscript.
We did revise the manuscript base on editor and reviewers' comments and suggestions all of the points and responses to the editor and reviewers of each point had been done and attached with this file.
Best Regards,
Sanun Jogloy
